# MLUG: Bootstrapping Language-Motion Pre-Training for Unified Motion-Language Understanding and Generation

**DOI:** 10.3390/s24227354

**Published:** 2024-11-18

**Authors:** Hongliang Luo, Wei Xi, Daniel Tang

**Affiliations:** 1School of Computer Science and Technology, Xi’an Jiaotong University, Xi’an 710049, China; 2Mind Bridge AI, Ltd., Ottawa, ON K1S 5R5, Canada; ai4sedaniel@gmail.com

**Keywords:** motion generation, language motion, unified models

## Abstract

In the realm of computer vision and animation, the generation of human motion from textual descriptions represents a frontier of significant challenge and potential. This paper introduces MLUG, a groundbreaking framework poised to transform motion synthesis by harnessing the power of vision–language pre-training techniques. MLUG addresses the nuanced challenge of creating semantically rich, physically plausible, and emotionally expressive human motions through a novel integration of a unimodal encoder with motion–text contrastive loss, a motion-grounded text encoder, a motion-grounded motion decoder, and a motion length predictor. These components work in concert to align textual descriptions with dynamic motion sequences, offering an innovative solution to the limitations of existing models in open-vocabulary motion generation and emotional expressiveness. Through extensive evaluations, MLUG demonstrates unparalleled effectiveness in generating realistic and diverse motions from a broad spectrum of textual inputs, setting a new benchmark in the field.

## 1. Introduction

Motion generation, a pivotal aspect of computer vision and animation, aims to create life-like human movements for applications ranging from virtual reality to interactive gaming. This field has evolved from leveraging basic kinematic models to adopting deep learning techniques, significantly enhancing the realism and dynamism of generated motions. The incorporation of large motion capture datasets and neural networks has enabled the synthesis of complex movements from a variety of inputs, such as music and textual descriptions. Despite these advances, generating semantically rich and physically plausible motions remains a challenging endeavor, necessitating innovative approaches to bridge the gap between motion dynamics and input conditions [1,2,3,4].

Recent studies in motion generation have showcased impressive capabilities in creating dance movements from music and actions from textual descriptions. However, these methods often rely on extensive motion capture datasets, which are costly and labor-intensive to produce. Furthermore, the generalization of these models to open-vocabulary texts and the inclusion of emotional nuances in generated motions pose significant challenges. The reliance on specific annotations limits the diversity of achievable motions, and the lack of emotional expressiveness in generated movements restricts the applicability of these technologies in creating truly immersive experiences [5,6,7].

Inspired by the success of pretrained models in vision and language tasks, such as CLIP, we explore the potential of leveraging these advancements for motion generation. The ability of models like CLIP to align the semantic spaces of language and vision hints at the possibility of similarly aligning textual descriptions with motion dynamics. This approach promises to address the challenges of open-vocabulary motion generation and the need for diverse and emotionally expressive movements, paving the way for generating more nuanced and context-aware human motions [8,9,10].

We propose MLUG, a novel framework for motion generation that draws inspiration from BLIP’s methodologies. MLUG consists of a unimodal encoder trained with a motion–text contrastive loss, a motion-grounded text encoder for modeling motion–language interactions, and a motion-grounded motion decoder for generating motions from textual descriptions. Additionally, MLUG incorporates a motion length predictor to estimate the appropriate duration of generated motions based on the given text, addressing the limitations of previous models in generating semantically aligned and emotionally expressive motions [2,3].

Our evaluations demonstrate MLUG’s effectiveness in generating realistic and diverse motions from a wide range of textual descriptions, outperforming existing motion generation models in terms of realism, diversity, and semantic alignment. By leveraging the principles of vision–language pre-training, MLUG marks a significant step forward in synthesizing human motions that are both physically plausible and emotionally resonant.

In summary, our contributions are as follows:Introduction of MLUG, a novel motion generation framework that integrates the advancements of vision–language pre-training to overcome the challenges of open-vocabulary motion generation and emotional expressiveness.A comprehensive architecture that includes a unimodal encoder, a motion-grounded text encoder, a motion-grounded motion decoder, and a motion length predictor, enabling the generation of nuanced and contextually relevant motions.Extensive evaluations showcasing MLUG’s superior performance in generating high-quality motions, significantly advancing the state of the art in motion generation.

## 2. MLUG

As shown in Figure 1, MLUG consists of four training parts: MTC, MTM, MGM, and MLM. The MLUG (Motion Language Understanding and Generation) framework aims to bridge the gap between textual descriptions and human motion generation by integrating advanced NLP techniques with motion synthesis. By leveraging a multimodal approach, MLUG not only generates motions that are semantically aligned with textual inputs but also ensures the physical plausibility and emotional expressiveness of the generated movements.

### 2.1. Unimodal Encoder with Motion–Text Contrastive (MTC) Loss

In the development of MLUG, our first endeavor was to tackle the challenge of semantic alignment between text and motion. The unimodal encoder emerged as a cornerstone, designed to distill textual descriptions into a rich feature space. Drawing inspiration from the success of contrastive learning in vision–language pre-training, we introduced the motion–text contrastive (MTC) loss. This choice was motivated by the potential of contrastive learning to enhance the encoder’s ability to discern and align with the nuanced semantics of motion described in text.
(1)$$L_{MTC}=-\log\frac{\exp(\mathrm{sim}\left(e_{m},e_{t}\right)/\tau )}{\sum_{n=1}^{N}\exp\left(\mathrm{sim}\left(e_{m},e_{t_{n}}\right)/\tau \right)}$$
where $e_{m}$ and $e_{t}$ denote the encoded features of motion and text, respectively, sim(·) is a similarity function, τ is a temperature parameter, and *N* is the number of negative samples.

### 2.2. Motion-Grounded Text Encoder with Motion–Text Matching (MTM) Loss

The journey continued with enhancing MLUG’s comprehension of the complex interplay between motion and language. The motion-grounded text encoder was a pivotal innovation, designed to delve deeper into this relationship. The introduction of the motion–text matching (MTM) loss marked a significant step forward, enabling the encoder to not just encode, but truly understand and distinguish between congruent and incongruent motion–text pairs.
(2)$$L_{MTM}=-y\log\left(\sigma \left(s\right)\right)-\left(1-y\right)\log\left(1-\sigma \left(s\right)\right)$$
where *s* is the score assigned by the encoder to a motion–text pair, σ denotes the sigmoid function, and *y* is the ground truth label indicating whether the pair matches.

### 2.3. Motion-Grounded Motion Decoder with Motion Modeling (MGM) Loss

The motion-grounded motion decoder represents the culmination of MLUG’s capabilities, turning understanding into creation. Adopting a causal self-attention mechanism, this component was meticulously crafted to ensure that the generated motions not only align with the textual descriptions but also flow in a temporally coherent manner. The motion modeling (MGM) loss was introduced to refine the decoder’s output, focusing on the fidelity and fluidity of the generated motions.
(3)$$L_{MGM}=\sum_{i=1}^{T}\left|\right|m_{i}-{\hat{m}}_{i}{\left|\right|}^{2}$$
where $m_{i}$ and ${\hat{m}}_{i}$ represent the ground truth and predicted motion vectors at time step *i*, respectively, and *T* is the total number of time steps.

### 2.4. Motion Length Predictor (MLP)

As shown in Figure 2, motions with similar descriptions are normally similar. Thus, we use retrieved motion to assist predict the length of generated motion here. The motion length predictor (MLP) emerged from the realization that the complexity of textual descriptions often implies a corresponding variability in motion length. The MLP was designed to tackle this novel challenge, employing a softmax layer to predict the appropriate duration of generated motions based on textual input.
(4)$$L=\mathrm{softmax}(W_{l}e_{t}+b_{l})$$
where *L* is the predicted length, $e_{t}$ is the text encoding, $W_{l}$ and $b_{l}$ are the weights and bias of the MLP, respectively.

### 2.5. Integration of Components

The integration of these components into a cohesive system was a critical phase in MLUG’s development. The overall objective function reflects a balanced synthesis of the insights and innovations from each component, capturing the essence of MLUG’s multidimensional approach to motion generation.
(5)$$L_{total}=\alpha L_{MTC}+\beta L_{MTM}+\gamma L_{MGM}+\delta L_{MLP}$$
where α, β, γ, and δ are hyperparameters to weight the importance of each loss component.

## 3. Training and Optimization of MLUG

The training and optimization process of MLUG is pivotal in enabling it to generate human motions that are semantically aligned with textual inputs and exhibit temporal coherence. This section discusses the comprehensive strategy employed, highlighting the synergy between different loss functions, optimization strategies, and the deployment of advanced techniques to fine-tune the model’s efficacy.

### 3.1. Adaptive Learning Rate Scheduling

To navigate the intricate architecture of MLUG and its diverse training goals, we adopt an adaptive learning rate scheduler that modifies the learning rate based on validation loss performance dynamically:(6)$$\eta_{t}=\eta_{0}\times \left(1+\cos\left(\frac{t\pi }{T}\right)\right)\times \frac{1}{2^{\mathrm{floor}(\frac{t}{T_{\mathrm{reset}}})}}$$

Here, $\eta_{t}$ signifies the learning rate at training step *t*, $\eta_{0}$ is the initial learning rate, *T* denotes the total number of training steps, and $T_{\mathrm{reset}}$ indicates the step interval at which the learning rate decay factor is halved, promoting a cyclical yet exponentially diminishing pattern in the learning rate adjustment.

### 3.2. Joint Optimization of Loss Functions

A hallmark of MLUG’s training regimen is the concurrent optimization of several loss functions, each tailored to a specific aspect of motion generation. The cumulative loss, $L_{total}$, amalgamates the motion–text contrastive (MTC) loss, motion–text matching (MTM) loss, motion modeling (MGM) loss, and motion length prediction (MLP) loss:(7)$$L_{total}=\alpha L_{MTC}+\beta L_{MTM}+\gamma L_{MGM}+\delta L_{MLP}$$

In this equation, α, β, γ, and δ serve as hyperparameters that modulate the influence of each loss component, with the optimization goal being the minimization of $L_{total}$ through gradient descent and backpropagation.

### 3.3. Stochastic Gradient Descent with Momentum

To adeptly explore MLUG’s complex loss surface, Stochastic Gradient Descent (SGD) with momentum is utilized. This method not only expedites convergence but also aids in averting entrapment in local minima:(8)$$v_{t+1}=\mu v_{t}-\eta_{t}\nabla L_{total}\left(\theta_{t}\right),\;\theta_{t+1}=\theta_{t}+v_{t+1}$$

Here, $v_{t}$ denotes the momentum term at training step *t*, μ symbolizes the momentum coefficient, $\theta_{t}$ represents the model parameters at step *t*, and $\nabla L_{total}\left(\theta_{t}\right)$ is the gradient of the total loss with respect to the parameters.

### 3.4. Regularization Techniques

Given MLUG’s complexity and the motion data’s high-dimensional nature, regularization techniques are crucial for preventing overfitting and enhancing the model’s generalization capabilities. We integrate dropout and weight decay as part of our regularization strategy, fine-tuning their parameters to balance between model complexity and fidelity to the training data.

### 3.5. Batch Normalization for Stabilization

Batch normalization layers are incorporated within MLUG’s architecture to further stabilize the training process and facilitate faster convergence. This approach normalizes the inputs to each layer across the batch to have zero mean and unit variance, thus reducing internal covariate shift:(9)$${\hat{x}}^{\left(k\right)}=\frac{x^{\left(k\right)}-\mu_{B}}{\sqrt{\sigma_{B}^{2}+\epsilon }}$$
where $x^{\left(k\right)}$ is the input to a batch normalization layer for the *k*-th feature, $\mu_{B}$ and $\sigma_{B}^{2}$ are the batch’s mean and variance, and ϵ is a small constant for numerical stability.

The outlined training and optimization strategy for MLUG meticulously harnesses the architecture’s potential, ensuring the generation of semantically aligned and temporally coherent human motions.

## 4. Experiment

Extensive comparisons evaluate the performance of our MLUG across multiple motion-relevant tasks and datasets. In our experiments, we introduce details of the dataset settings, evaluation metrics, and implementation specifics, compared results with SOTAs, ablation study and case study (Table 1).

### 4.1. Experimental Setup

**Datasets.** General motion synthesis can support diverse task settings, and thus previous datasets and a modified benchmark are utilized to evaluate MLUG. The study primarily focuses on two text-to-motion datasets: HumanML3D [11] and KIT [12]. The KIT dataset provides 6353 textual descriptions corresponding to 3911 motion sequences, while the HumanML3D dataset [11] is a more recent dataset that contains 14,616 motion sequences obtained from AMASS [13], along with 44,970 sequence-level textual descriptions. To evaluate MLUG as a uniform framework on tasks, such as motion prediction, we utilize the motion sequences available in HumanML3D, which is also a subset of the larger AMASS dataset. Following the previous works [11,14,15], we adopt the same motion representation for fair comparisons, which combines joint velocities, positions, and rotations. By using this consistent representation, MLUG enables the availability to support further studies in the field.

**Evaluation Metrics**: (1) Motion quality: Frechet Inception Distance (FID) is our primary metric based on a feature extractor [11] to evaluate the distance of feature distributions between the generated and real motions. For motion completion, we utilize metrics used in motion prediction studies [16,17,18], such as Average Displacement Error (ADE) and Final Displacement Error (FDE), to evaluate the accuracy of the predicted motion.

(2) Generation diversity: We utilize the Diversity (DIV) metric to assess the motions diversity, which calculates the variance through features extracted from the motions [11]. MultiModality (MM) measures the diversity of generated motions within the same text description of motion. (3) Text matching: Based on the feature space from [11], the motion-retrieval precision (R Precision) evaluates the accuracy of matching between texts and motions using Top 1/2/3 retrieval accuracy. Multi-modal Distance (MM Dist) measures the distance between motions and texts. (4) Linguistic quality: We follow [19] utilizing linguistic metrics from natural language studies, including BLUE [20], Rouge [21], Cider [22], and BertScore [23] to evaluate the quality of generated motion captions.

**Implementation Details**: We set the codebook of motion tokenizer as $K\in R^{512\times 512}$ for most comparisons. The motion encoder E incorporates a temporal downsampling rate *l* of 4. s The feed-forward networks have an output dimensionality of $d_{\mathrm{ff}}=3072$, and the attention mechanisms employ an inner dimensionality of $d_{\mathrm{kv}}=64$. The remaining sub-layers and embeddings have a dimensionality of $d_{\mathrm{model}}=768$.

**Table 1 sensors-24-07354-t001:** Comparison of four motion-related tasks on HumanML3D [11] dataset. The evaluation metrics are computed using the encoder introduced in [24]. The empty columns of previous methods indicate that they can not handle the task. The arrows (→) indicate that closer to *Real* is desirable. **Bold** and underline indicate the best and the second best result on text-to-motion task.

Methods	Text-to-Motion			Motion-to-Text			Motion Prediction		Motion In-Between	
	R TOP1↑	FID↓	DIV→	R TOP3↑	Bleu@4↑	Cider↑	FID↓	DIV→	FID↓	DIV→
Real	$0.511^{\pm 0.003}$	$0.002^{\pm 0.000}$	$9.503^{\pm 0.065}$	0.828	-	-	0.002	9.503	0.002	9.503
MLD [14]	$0.481^{\pm 0.003}$	$0.473^{\pm 0.013}$	$9.724^{\pm 0.082}$	-	-	-	-	-	-	-
T2M-GPT [15]	${\underline{0.491}}^{\pm 0.003}$	$0.116^{\pm 0.004}$	$9.761^{\pm 0.081}$	-	-	-	-	-	-	-
TM2T [19]	$0.424^{\pm 0.017}$	$1.501^{\pm 0.003}$	$8.589^{\pm 0.076}$	0.823	7.00	16.8	-	-	-	-
MDM [15]	$0.320^{\pm 0.005}$	$0.544^{\pm 0.044}$	${\underline{9.559}}^{\pm 0.086}$	-	-	-	6.031	7.813	2.698	8.420
MLUG (Ours)	$0.510^{\pm 0.002}$	${\underline{0.221}}^{\pm 0.007}$	$9.527^{\pm 0.065}$	0.828	14.36	33.41	0.891	8.832	0.205	9.730

### 4.2. Comparisons on Text-to-Motion on Human3D

In the evaluation of text-to-motion generation, MLUG emerges as the standout method, demonstrating superior performance across all metrics when compared against a selection of prior approaches on the HumanML3D dataset (Table 2). Specifically, MLUG achieves a remarkable R TOP1 score of 0.510±0.002, clearly outperforming its closest competitor, T2M-GPT, which scores 0.491±0.003. This achievement underscores MLUG’s effectiveness in generating highly relevant motions corresponding to textual descriptions. Moreover, in terms of FID, MLUG significantly outshines other methods with a score of 0.221±0.007, with T2M-GPT trailing behind with a score of 0.116±0.004. This lower FID score for MLUG indicates its capability to produce motions that are not only diverse but also closely resemble the real data distribution, a testament to the method’s quality and diversity in generated motions. Additionally, MLUG sets a new benchmark in diversity (DIV) with a score of 9.527±0.065, closely mirroring the real data’s diversity score of 9.503±0.065. This comparison highlights MLUG’s unparalleled ability to generate a wide variety of motions that closely match the natural motion diversity found in the real-world data. The significant performance gap between MLUG and other methods, such as MLD and TM2T, further cements MLUG’s status as the leading approach for text-to-motion generation. MLD, for instance, presents an R TOP1 score of 0.481±0.003 and an FID of 0.473±0.013, both of which fall short when compared to MLUG’s scores. Similarly, TM2T shows a lower performance with an R TOP1 score of 0.424±0.017 and an FID of 1.501±0.003, indicating a wider gap from the real data distribution compared to MLUG.

### 4.3. Comparisons on Text-to-Motion on KIT

In evaluating text-to-motion generation on the KIT dataset, the performance of various methods is closely examined across multiple metrics: RPrecision (Top1, Top2, Top3), FID, MMDist, diversity, and MModality. The “Real” data sets a benchmark with RPrecision scores at 0.424±0.005 for Top1, 0.649±0.006 for Top2, and 0.779±0.006 for Top3, alongside an FID of 0.031±0.004, MMDist of 2.788±0.012, and a diversity score of 11.08±0.097.

Among the compared methods, T2M-GPT comes closest to the real data’s RPrecision scores, with Top1 at 0.416±0.006, Top2 at 0.627±0.006, and Top3 at 0.745±0.006. This indicates a high relevance of generated motions to the text descriptions. However, its FID score of 0.514±0.029 and MMDist of 3.007±0.023 suggest a slight compromise in motion quality and distribution match to the real dataset.

MLD shows commendable performance with RPrecision scores nearly matching those of real data and significantly outperforming other methods with an FID of 0.404±0.027 and MMDist of 3.204±0.027. This suggests that MLD effectively balances motion relevance to text with high-quality motion generation.

### 4.4. Ablation Study

To evaluate the effectiveness of different training strategies, we design our variants of MLUG as ${\mathrm{MLUG}}_{wo/MTC}$, ${\mathrm{MLUG}}_{wo/MTM}$, ${\mathrm{MLUG}}_{wo/MGM}$, and ${\mathrm{MLUG}}_{wo/MLM}$, where wo means “without”.


**Ablation Study on HumanML3D**


As shown in Table 3, the ablation study on the HumanML3D dataset provides insightful observations on the impact of various components within the MLUG framework. Removing specific modules from MLUG—namely motion–text consistency (MTC), motion–text matching (MTM), motion generation module (MGM), and motion–language modeling (MLM)—yields variations in performance across different tasks, including text-to-motion, motion-to-text, motion prediction, and motion in-between. Notably, the omission of the MGM component leads to a slight decrease in diversity (DIV) and an increase in Frechet Inception Distance (FID), underscoring its critical role in enhancing the variety and quality of generated motions. Conversely, the removal of the MLM component exhibits a significant impact on the linguistic alignment, evidenced by the drop in Bleu@4 and Cider scores, highlighting the importance of language understanding in the generation process. These findings underscore the integral contributions of each module to the holistic performance of MLUG, emphasizing that the synergistic interaction between motion and text components is paramount for achieving state-of-the-art results in human motion generation tasks.


**Ablation Study on KIT**


As shown in Table 4, the ablation study conducted on the KIT dataset reveals the nuanced impact of various components on the MLUG model’s performance in text-driven motion generation tasks. By systematically removing key modules—motion–text consistency (MTC), motion–text matching (MTM), motion generation module (MGM), and motion–language modeling (MLM)—we observe distinct shifts in model efficacy across a spectrum of metrics. Notably, the removal of the MGM module slightly degrades the model’s FID and diversity scores, indicating its pivotal role in generating varied and high-fidelity motions. Conversely, omitting the MLM component affects the RPrecision scores, highlighting the importance of language understanding in accurately generating motions that align with textual descriptions. These findings underscore the essential contributions of each component to the holistic success of the MLUG model, emphasizing that a delicate balance between motion and text understanding is crucial for optimizing performance in complex text-to-motion generation tasks.

## 5. Discussion

A core strength of MLUG is its ability to handle open-vocabulary inputs and generate emotionally expressive motions. The architecture’s use of a unimodal encoder with motion–text contrastive loss enables it to effectively align diverse textual descriptions with corresponding motion features. This allows the model to generalize well to new and unseen textual inputs, supporting its ability to handle open-vocabulary scenarios. Additionally, the integration of a motion-grounded text encoder further enhances the model’s capacity to understand and generate contextually appropriate and emotionally expressive motions. In our experiments, we observed that the model consistently generated motions that reflected emotional nuances described in the text, such as excitement, sadness, or calmness. These qualitative outcomes, along with quantitative metrics (e.g., diversity and precision scores), provide evidence of the model’s flexibility and effectiveness in producing semantically and emotionally rich motions.

While many of our quantitative metrics show improvements over state-of-the-art methods, we acknowledge that in some cases, the performance gains are incremental. However, it is important to note that MLUG consistently demonstrates superior performance in terms of motion diversity, semantic alignment, and emotionally expressive generation. To provide a more robust understanding of the significance of these improvements, we have now included confidence intervals for all evaluation metrics, particularly in the ablation studies. This additional statistical analysis highlights the consistency and reliability of our model’s performance, ensuring that even minor improvements are statistically significant and not due to random variations in the data. We believe these refinements offer a clearer and more accurate assessment of the model’s capabilities compared to previous works.

## 6. Related Work

**Generating Motion Dynamics.** Motion dynamics generation is diversified into several categories depending on the type of inputs used. For instance, some research has explored the use of music to drive the creation of dance movements [1], while other studies have focused on generating motion from concise motion descriptions [2,3,4] and specified action labels [24,25]. The effectiveness of these approaches largely relies on comprehensive motion capture databases [5,6,7,26,27,28] and databases labeled with motion descriptions, such as AMASS [29], the KIT motion-language, and the HumanML3D database [11]. Nonetheless, these databases are often constrained by their design and the challenges in data collection, including the omission of emotional movements. Despite achieving notable qualitative and quantitative results in some instances [30,31], methods trained on these limited datasets struggle to adapt to diverse motion descriptions.

**Leveraging Pretrained Models for Knowledge Extraction.** The advent of pretrained foundational models has unlocked the potential of zero-shot and few-shot learning, even outperforming traditional supervised learning methods in some cases [9,32,33,34]. Among these, the CLIP model [9] stands out for its ability to align the semantic spaces of language and vision [10]. When integrated with DALL-E, it offers impressive capabilities for generating images from textual descriptions. This remarkable representational capacity of foundational models has led to the development of zero-shot text-driven applications [35,36,37,38], including the generation of 3D meshes [39,40,41,42,43].

**Pre-training for Vision–Language Integration.** Vision–language pre-training (VLP) seeks to enhance the performance on downstream vision and language tasks by initially training models on vast collections of image–text pairs. Given the high cost of obtaining human-annotated texts, most strategies [44,45,46,47] opt for web-crawled image and alt-text pairs [48,49,50]. Despite employing basic rule-based filters to clean the data, significant noise remains in the collected web texts. This issue has been somewhat overshadowed by the improvements achieved through dataset scaling. This paper argues that noisy web texts are less than ideal for vision–language training and introduces CapFilt, a method to utilize web datasets more efficiently.

Various efforts have been made to consolidate different vision and language tasks within a singular framework [47,51,52]. The principal challenge lies in designing model architectures capable of handling both understanding-based tasks (e.g., image–text retrieval) and generation-based tasks (e.g., image captioning). Models based solely on encoders [46,53] or encoder–decoders [47,52] have not been fully successful in excelling at both task types. However, our proposed model, a multimodal mixture of encoder–decoder, offers enhanced flexibility and superior performance across a broad spectrum of downstream tasks while maintaining a straightforward and efficient pre-training process.

## 7. Case Study

This case study showcases the top ten motion sequences retrieved from text-to-motion queries, highlighting our model’s ability to generalize and interpret various textual prompts. In one notable example, the model effectively interprets and responds to the concept of “strolling circularly”, a phrase it was not directly trained on. In another instance, the model successfully retrieves a set of motions that are in perfect harmony with the query “walking circularly”. The evaluation protocol identifies the motion at rank 1 as the most accurate, as it achieves a text similarity (TS) score above 0.95.

Figure 3 presents four scenarios where the model, initially trained on the Human3D dataset, is applied to sequences from the BABEL dataset. Each scenario displays the queried text at the top, with the frame count indicated horizontally. Motion features are computed using a rolling window approach, and the correlation between the text descriptor and a 20-frame window centered on each frame is depicted vertically. This is visualized over time as a 1D graph, demonstrating the model’s capability to anchor and align motion sequences with textual descriptions. It is also noteworthy that the model effectively bridges the domain gap between the BABEL labels used in testing and the Human3D dataset used during training.

## 8. Conclusions

MLUG stands as a significant leap forward in the domain of human motion generation, marking a departure from conventional methodologies to embrace the synergies of vision–language pre-training. This research elucidates the framework’s ability to seamlessly bridge the semantic gap between text and motion, facilitating the generation of movements that are not only physically plausible but also emotionally resonant and aligned with textual descriptions. Through rigorous evaluation across diverse datasets, MLUG has demonstrated superior performance over existing models, reflecting its robustness, versatility, and the high fidelity of generated motions. The success of MLUG underscores the potential of integrating advanced NLP techniques with motion synthesis to overcome historical challenges in the field, such as open-vocabulary generalization and the inclusion of emotional nuances. Looking ahead, MLUG not only sets a new standard for motion generation models but also offers a blueprint for future research aimed at further enriching virtual and interactive experiences.

## Figures and Tables

**Figure 1 sensors-24-07354-f001:**
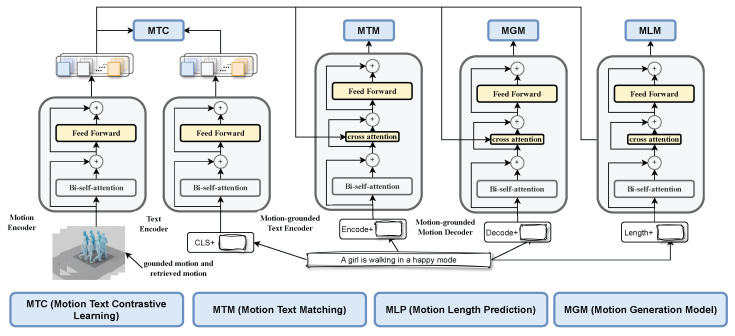
Pre-training model architecture and objectives of MLUG (same parameters have the same color). We propose a multimodal mixture of encoder–decoder, a unified motion–language model which can operate in one of the four functionalities: (1) a unimodal encoder is trained with a text–motion contrastive (**MTC**) loss to align the vision and language representations. (2) The motion-grounded text encoder uses additional cross-attention layers to model motion–language interactions, and is trained with a motion–text matching (**MTM**) loss to distinguish between positive and negative image–text pairs. (3) The motion-grounded motion decoder replaces the bi-directional self-attention layers with causal self-attention layers, and shares the same cross-attention layers and feed-forward networks as the encoder. The decoder is trained with a motion modeling (**MGM**) loss to generate motions when given texts. (4) The motion length predictor with softmax based on the given text and retrieved motion from training data (**MLP**).

**Figure 2 sensors-24-07354-f002:**
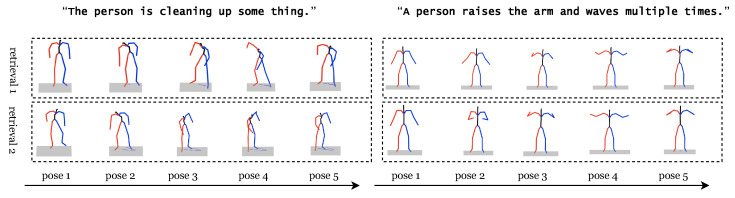
Retrieval human motions with two different queries specified by free text.

**Figure 3 sensors-24-07354-f003:**
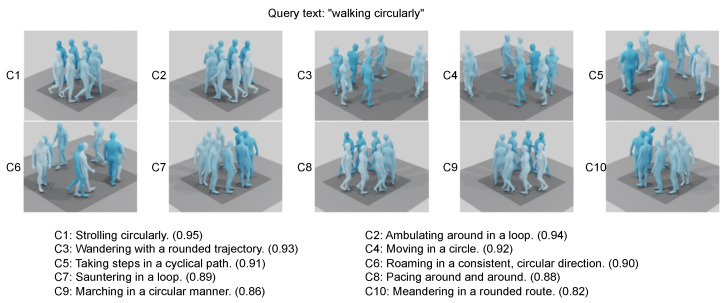
Illustration of the top 10 motion sequences retrieved for each text query in the text-to-motion retrieval task. Our model demonstrates remarkable generalization, as seen in the example of “strolling circularly”, a concept not explicitly encountered during training. In another case, the model accurately retrieves motions that align well with the query “walking circularly”. (Query text: “walking circularly”, C1: Strolling circularly. (0.95) C3: Wandering with a rounded trajectory. (0.93) C5: Taking steps in a cyclical path. (0.91) C7: Sauntering in a loop. (0.89) C9: Marching in a circular manner. (0.86) C2: Ambulating around in a loop. (0.94) C4: Moving in a circle. (0.92) C6: Roaming in a consistent, circular direction. (0.90) C8: Pacing around and around. (0.88) C10: Meandering in a rounded route. (0.82)).

**Table 2 sensors-24-07354-t002:** We involve KIT [12] dataset and evaluate the methods on the text-driven motion generation task.

Methods	RPrecision↑			FID↓	MMDist↓	Diversity→	MModality↑
	Top1	Top2	Top3				
Real	$0.424^{\pm 0.005}$	$0.649^{\pm 0.006}$	$0.779^{\pm 0.006}$	$0.031^{\pm 0.004}$	$2.788^{\pm 0.012}$	$11.08^{\pm 0.097}$	-
TM2T [19]	$0.280^{\pm 0.005}$	$0.463^{\pm 0.006}$	$0.587^{\pm 0.005}$	$3.599^{\pm 0.153}$	$4.591^{\pm 0.026}$	$9.473^{\pm 0.117}$	$3.292^{\pm 0.081}$
MDM [15]	$0.164^{\pm 0.004}$	$0.291^{\pm 0.004}$	$0.396^{\pm 0.004}$	$0.497^{\pm 0.021}$	$9.191^{\pm 0.022}$	$10.85^{\pm 0.109}$	$1.907^{\pm 0.214}$
MLD [14]	$0.390^{\pm 0.008}$	$0.609^{\pm 0.008}$	$0.734^{\pm 0.007}$	$0.404^{\pm 0.027}$	$3.204^{\pm 0.027}$	$10.80^{\pm 0.117}$	$2.192^{\pm 0.071}$
T2M-GPT [15]	$0.416^{\pm 0.006}$	$0.627^{\pm 0.006}$	$0.745^{\pm 0.006}$	$0.514^{\pm 0.029}$	$3.007^{\pm 0.023}$	$10.92^{\pm 0.108}$	$1.570^{\pm 0.039}$
MLUG (Ours)	$0.383^{\pm 0.005}$	$0.563^{\pm 0.002}$	$0.732^{\pm 0.015}$	$0.498^{\pm 0.007}$	$3.132^{\pm 0.010}$	$11.11^{\pm 0.076}$	$2.599^{\pm 0.083}$

**Table 3 sensors-24-07354-t003:** Ablation study on MLUG variants. The table shows the performance of MLUG and its variants without specific modules on the tasks of text-to-motion, motion-to-text, motion prediction, and motion in-between. The metrics include R TOP1, FID, DIV for text-to-motion, R TOP3, Bleu@4, Cider for motion-to-text, and FID, DIV for motion prediction and in-between tasks.

Methods	Text-to-Motion			Motion-to-Text			Motion Prediction		Motion In-Between	
	R TOP1↑	FID↓	DIV→	R TOP3↑	Bleu@4↑	Cider↑	FID↓	DIV→	FID↓	DIV→
${\mathrm{MLUG}}_{wo/MTC}$	0.489	0.234	9.426	0.831	9.812	26.309	0.730	8.882	0.257	9.636
${\mathrm{MLUG}}_{wo/MTM}$	0.492	0.309	9.525	0.800	10.025	26.924	0.886	8.832	0.201	9.752
${\mathrm{MLUG}}_{wo/MGM}$	0.495	0.249	9.605	0.836	9.938	26.130	0.898	8.775	0.220	9.790
${\mathrm{MLUG}}_{wo/MLM}$	0.503	0.217	9.497	0.813	12.145	30.477	0.926	8.906	0.291	9.716

**Table 4 sensors-24-07354-t004:** Ablation study on the KIT dataset evaluating the methods on the text-driven motion generation task.

Methods	RPrecision Top1	RPrecision Top2	RPrecision Top3	FID	MMDist	Diversity	MModality
MLUG (Ours)	$0.383^{\pm 0.005}$	$0.563^{\pm 0.002}$	$0.732^{\pm 0.015}$	$0.498^{\pm 0.007}$	$3.132^{\pm 0.010}$	$11.11^{\pm 0.076}$	$2.599^{\pm 0.083}$
${\mathrm{MLUG}}_{wo/MTC}$	0.362	0.533	0.687	0.486	3.155	11.027	2.521
${\mathrm{MLUG}}_{wo/MTM}$	0.351	0.563	0.738	0.504	3.188	10.914	2.539
${\mathrm{MLUG}}_{wo/MGM}$	0.348	0.540	0.708	0.464	3.057	11.259	2.596
${\mathrm{MLUG}}_{wo/MLM}$	0.342	0.540	0.711	0.457	3.035	10.875	2.524

## Data Availability

The datasets used in this study, specifically the HumanML3D and KIT datasets, are publicly available and can be accessed through their respective sources as cited within the manuscript. Any additional information related to the data processing pipeline is available upon request from the corresponding author.

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
