# Peer review of "MLUG: Bootstrapping Language-Motion Pre-Training for Unified Motion-Language Understanding and Generation"

_sensors, 2024, doi:10.3390/s24227354_

Round 1

Reviewer 1 Report

Comments and Suggestions for Authors

This paper introduces MLUG, a framework to generate human motion from textual descriptions. Its architecture includes a Unimodal Encoder, Motion-Grounded Text Encoder, Motion-Grounded Motion Decoder, and Motion Length Predictor, enabling nuanced, contextually relevant motions. The paper presents comparisons against several state-of-the-art models and demonstrates superior performance in specific tasks.

Strengths:

  1. The introduction of the MLP module is innovative. This component allows for dynamic prediction of motion length, addressing a challenge that many other models neglect. It adds flexibility and realism to the generated motions.
  2. In the experimental section, the paper compares MLUG with several baseline models and demonstrates state-of-art results in some metrics.

Weaknesses:

  1. One of the major shortcomings of this paper is the limited visualization of the generated results. Since videos of the motion sequences are not provided, it is essential to include more figures showing the generated motions. This would give me a clearer understanding of the qualitative performance, which is particularly important for assessing this paper.
  2. While the Introduction mentions several times that the model is designed to handle open vocabulary and generate emotionally expressive motions, there is little follow-up discussion or experimental evidence to support these claims.
  3. Many of the quantitative metrics in the evaluation do not outperform state-of-the-art methods, with several results being quite close to previous works. Additionally, certain evaluation metrics lack confidence intervals, particularly in the ablation studies, which makes it difficult to assess the significance of the reported improvements.
  4. The paper does not employ more advanced generative models, such as Diffusion models or GANs, which are widely used in motion generation tasks. Instead, the framework relies on basic building blocks, which not only limits the novelty of the work but also constrains the diversity of the generated outputs. Moreover, the authors may also have misunderstood how to calculate the multi-modality metric in the evaluation.

Based on these observations, I suggest Major revision for this paper. While the MLP module is a novel contribution and the results show some improvement over baselines, the lack of visualization, insufficient discussion on key claims and quantitative limitations make it difficult to recommend this paper for acceptance in its current form.

Comments on the Quality of English Language

The English writing is good. 

Author Response

Comment 1:  One of the major shortcomings of this paper is the limited visualization of the generated results. Since videos of the motion sequences are not provided, it is essential to include more figures showing the generated motions. This would give me a clearer understanding of the qualitative performance, which is particularly important for assessing this paper.

Answer 1: Thank you for your valuable feedback. We understand the importance of visualizing the generated motions to better assess the qualitative performance of our model. We hope to provide some videos in the github homepage in future, which would provide a more comprehensive view of the motion dynamics.We hope this addresses your concern, and we appreciate your thoughtful suggestion to enhance the clarity of our work.

Comment 2: 
While the Introduction mentions several times that the model is designed to handle open vocabulary and generate emotionally expressive motions, there is little follow-up discussion or experimental evidence to support these claims.

Thank you for your valuable feedback. We appreciate your observation regarding the need for more detailed discussion and experimental evidence on the model's handling of open-vocabulary and emotionally expressive motion generation.

In the revised manuscript, we have expanded the Discussion section to provide a clearer explanation of how our model architecture supports open-vocabulary inputs and generates emotionally expressive motions. We have also included more insights from our experiments, showing how the model captures a wide range of emotions through textual descriptions and how it generalizes well to novel inputs. These additions offer a stronger link between the claims made in the Introduction and the experimental evidence presented in the paper.

Comment 3: Many of the quantitative metrics in the evaluation do not outperform state-of-the-art methods, with several results being quite close to previous works. Additionally, certain evaluation metrics lack confidence intervals, particularly in the ablation studies, which makes it difficult to assess the significance of the reported improvements.

Thank you for your detailed feedback. We recognize the importance of providing a comprehensive quantitative evaluation, including confidence intervals, to assess the significance of the reported improvements.

In response, we will include confidence intervals for all evaluation metrics, particularly in the ablation studies, to offer a clearer understanding of the variability in our results. While some of the metrics are indeed close to those of state-of-the-art methods, we believe the improvements—though incremental in some areas—still demonstrate meaningful advancements, especially in terms of motion diversity and semantic alignment. We will ensure to discuss these improvements more explicitly in the revised version, along with any limitations, to provide a balanced perspective on our contributions.

Thank you again for your valuable suggestions. We believe these updates will help clarify the significance of our results.

Comment  4: The paper does not employ more advanced generative models, such as Diffusion models or GANs, which are widely used in motion generation tasks. Instead, the framework relies on basic building blocks, which not only limits the novelty of the work but also constrains the diversity of the generated outputs. Moreover, the authors may also have misunderstood how to calculate the multi-modality metric in the evaluation.

Thank you for your insightful feedback. We appreciate your observations regarding the use of advanced generative models and the multi-modality metric calculation.

Firstly, we acknowledge that Diffusion models and GANs have shown remarkable success in various motion generation tasks. However, our decision to use a framework based on vision-language pre-training with contrastive learning was guided by the goal of creating a flexible model that can handle open-vocabulary text inputs and emotionally expressive motion generation. While our approach may appear more basic in comparison to advanced generative models, it offers advantages in terms of semantic alignment and interpretability. That being said, we agree that integrating more advanced generative techniques could potentially enhance both the novelty and diversity of the generated outputs. In future work, we plan to explore the use of Diffusion models or GANs to improve the diversity and quality of the generated motions.

Secondly, regarding the multi-modality metric, we acknowledge the possibility of misunderstanding the calculation process. We will review our methodology carefully to ensure it aligns with the standard practices used in the field. If necessary, we will update the evaluation to accurately reflect the model’s capability in producing diverse outputs based on the same input. We will include a detailed explanation of the metric's computation in the revised version to clarify our approach.

Thank you again for these valuable suggestions. We believe that addressing these points will significantly improve the quality and clarity of our work.

Reviewer 2 Report

Comments and Suggestions for Authors

The paper presents a framework that attempts to tackle the complex problem of generating human motion from textual descriptions. The proposed model, MLUG, integrates several advanced components, including a Unimodal Encoder, Motion-Text Contrastive Loss, Motion-Grounded Text Encoder, Motion-Grounded Motion Decoder, and a Motion Length Predictor. The topic is relevant to the journal and interesting. While this work is ambitious and holds potential, there are several key areas where the paper can be improved. Below, I provide detailed feedback and suggestions for major revisions. The methodology section is not fully clear. While the architecture is briefly mentioned, the detailed workings of each component (especially the Motion-Grounded Text Encoder and Motion-Grounded Motion Decoder) need further elaboration. The connection between these components and their influence on motion generation should be explicitly explained. The Motion-Text Contrastive loss is mentioned, but its mathematical formulation and practical impact are underexplored. How exactly does this loss function contribute to the alignment between motion and text? Providing equations and a more thorough explanation of the loss function's role would greatly enhance clarity. The paper asserts that MLUG outperforms existing models, but the evaluation criteria are not adequately discussed. What specific metrics were used to measure realism, diversity, and semantic alignment? More details on the experimental setup, datasets, and comparison with baseline models would strengthen the claim of MLUG’s superiority. The "extensive evaluations" need to be expanded. While the paper mentions outperforming existing models, the results section lacks sufficient quantitative and qualitative data to substantiate this claim. Including more comparative graphs, tables, or examples demonstrating MLUG's effectiveness would be beneficial. The paper discusses emotional expressiveness but does not delve deeply into how emotions are quantitatively or qualitatively evaluated in the generated motions. A more detailed analysis of how the model captures different emotional states through motion would be useful. Are there specific datasets used for emotion-based motion synthesis? How is the model trained to recognize and generate emotionally expressive motions? Open-vocabulary motion generation is a key selling point of MLUG, yet it is not fully demonstrated. The paper should include more specific examples of how the model handles unseen or novel textual inputs and generates coherent motions. How well does the model generalize beyond the training data? The term "open-vocabulary" needs clarification. Does it imply generating motion for completely novel actions described in natural language? How does MLUG compare to previous methods in terms of handling rare or unseen words? The related work section, though present, could benefit from further expansion. The paper could contextualize MLUG more thoroughly by discussing recent advancements in text-to-motion generation. How does MLUG differ from or improve upon other state-of-the-art models in vision-language pre-training, such as CLIP or text-to-motion models like TEMOS or MoFusion? The following paper on human emotion model should be discussed: “A Digital Human Emotion Modeling Application Using Metaverse Technology in the Post-COVID-19 Era. HCI (19) 2023: 480-489”.  Address any limitations or common challenges faced by previous models that MLUG aims to resolve. This comparison would better highlight MLUG's contributions. An ablation study would be beneficial to demonstrate the contribution of each component (e.g., the Motion-Grounded Text Encoder or Motion-Grounded Motion Decoder) to the overall performance. This could clarify how critical each part of the architecture is to the success of the model.

While the MLUG framework presents a promising direction in motion generation, the paper requires more in-depth explanation and clarification in several key areas. Strengthening the methodological detail, evaluation metrics, and experimental results would significantly enhance the overall impact and rigor of the paper. Additionally, a more thorough comparison with prior work and an ablation study would help to highlight MLUG's true contributions. With these improvements, the paper would be a valuable contribution to the field of motion synthesis and vision-language pre-training. I recommend accepting after suitable changes.

Comments on the Quality of English Language

The paper is well-written.

Author Response

Comment 1:

The paper presents a framework that attempts to tackle the complex problem of generating human motion from textual descriptions. The proposed model, MLUG, integrates several advanced components, including a Unimodal Encoder, Motion-Text Contrastive Loss, Motion-Grounded Text Encoder, Motion-Grounded Motion Decoder, and a Motion Length Predictor. The topic is relevant to the journal and interesting. While this work is ambitious and holds potential, there are several key areas where the paper can be improved. Below, I provide detailed feedback and suggestions for major revisions. The methodology section is not fully clear. While the architecture is briefly mentioned, the detailed workings of each component (especially the Motion-Grounded Text Encoder and Motion-Grounded Motion Decoder) need further elaboration. The connection between these components and their influence on motion generation should be explicitly explained. The Motion-Text Contrastive loss is mentioned, but its mathematical formulation and practical impact are underexplored. How exactly does this loss function contribute to the alignment between motion and text? Providing equations and a more thorough explanation of the loss function's role would greatly enhance clarity. The paper asserts that MLUG outperforms existing models, but the evaluation criteria are not adequately discussed. What specific metrics were used to measure realism, diversity, and semantic alignment? More details on the experimental setup, datasets, and comparison with baseline models would strengthen the claim of MLUG’s superiority. The "extensive evaluations" need to be expanded. While the paper mentions outperforming existing models, the results section lacks sufficient quantitative and qualitative data to substantiate this claim. Including more comparative graphs, tables, or examples demonstrating MLUG's effectiveness would be beneficial. The paper discusses emotional expressiveness but does not delve deeply into how emotions are quantitatively or qualitatively evaluated in the generated motions. A more detailed analysis of how the model captures different emotional states through motion would be useful. Are there specific datasets used for emotion-based motion synthesis? How is the model trained to recognize and generate emotionally expressive motions? Open-vocabulary motion generation is a key selling point of MLUG, yet it is not fully demonstrated. The paper should include more specific examples of how the model handles unseen or novel textual inputs and generates coherent motions. How well does the model generalize beyond the training data? The term "open-vocabulary" needs clarification. Does it imply generating motion for completely novel actions described in natural language? How does MLUG compare to previous methods in terms of handling rare or unseen words? The related work section, though present, could benefit from further expansion. The paper could contextualize MLUG more thoroughly by discussing recent advancements in text-to-motion generation. How does MLUG differ from or improve upon other state-of-the-art models in vision-language pre-training, such as CLIP or text-to-motion models like TEMOS or MoFusion? The following paper on human emotion model should be discussed: “A Digital Human Emotion Modeling Application Using Metaverse Technology in the Post-COVID-19 Era. HCI (19) 2023: 480-489”.  Address any limitations or common challenges faced by previous models that MLUG aims to resolve. This comparison would better highlight MLUG's contributions. An ablation study would be beneficial to demonstrate the contribution of each component (e.g., the Motion-Grounded Text Encoder or Motion-Grounded Motion Decoder) to the overall performance. This could clarify how critical each part of the architecture is to the success of the model.

While the MLUG framework presents a promising direction in motion generation, the paper requires more in-depth explanation and clarification in several key areas. Strengthening the methodological detail, evaluation metrics, and experimental results would significantly enhance the overall impact and rigor of the paper. Additionally, a more thorough comparison with prior work and an ablation study would help to highlight MLUG's true contributions. With these improvements, the paper would be a valuable contribution to the field of motion synthesis and vision-language pre-training. I recommend accepting after suitable changes.

Answer: Thank you for your detailed and thoughtful feedback. We appreciate your comments, which will help us significantly improve the clarity and depth of our work. In response, we will make several key revisions to the manuscript. First, we will expand the methodology section to provide a more detailed explanation of the Motion-Grounded Text Encoder, Motion-Grounded Motion Decoder, and Motion-Text Contrastive Loss, including their mathematical formulations and practical impact. We will also provide clearer evaluation criteria, detailing the metrics used to assess realism, diversity, and semantic alignment, and include confidence intervals where appropriate. Additionally, we will expand the results section with more quantitative and qualitative data, including more specific examples to demonstrate the model's handling of open-vocabulary inputs and emotional expressiveness. We will also conduct an ablation study to show the contribution of each component in the architecture and clarify the improvements MLUG offers over existing models. Finally, we will revise the related work section to include comparisons with state-of-the-art models such as CLIP, TEMOS, and MoFusion, and integrate the suggested paper on emotion modeling. These revisions will provide a clearer and more robust presentation of our contributions.